# Crucial Role of FABP3 in αSyn-Induced Reduction of Septal GABAergic Neurons and Cognitive Decline in Mice

**DOI:** 10.3390/ijms22010400

**Published:** 2021-01-01

**Authors:** Kazuya Matsuo, Yasushi Yabuki, Ronald Melki, Luc Bousset, Yuji Owada, Kohji Fukunaga

**Affiliations:** 1Department of Pharmacology, Graduate School of Pharmaceutical Sciences, Tohoku University, Sendai 980-8578, Japan; kazuya.matsuo.q8@dc.tohoku.ac.jp (K.M.); yabukiy@kumamoto-u.ac.jp (Y.Y.); 2Department of Genomic Neurology, Institute of Molecular Embryology and Genetics, Kumamoto University, Kumamoto 860-0811, Japan; 3Laboratory of Neurodegenerative Diseases, CEA, Institut François Jacob (MIRcen) and CNRS, 18 Route du Panorama, 92265 Fontenay-aux-Roses, France; ronald.melki@cnrs.fr (R.M.); Luc.BOUSSET@cnrs.fr (L.B.); 4Department of Organ Anatomy, Graduate School of Medicine, Tohoku University, Sendai 980-0872, Japan; owada@med.tohoku.ac.jp

**Keywords:** α-synuclein, cognition, fatty acid-binding protein, gamma-aminobutyric acid, medial septum

## Abstract

In synucleinopathies, while motor symptoms are thought to be attributed to the accumulation of misfolded α-synuclein (αSyn) in nigral dopaminergic neurons, it remains to be elucidated how cognitive decline arises. Here, we investigated the effects of distinct αSyn strains on cognition and the related neuropathology in the medial septum/diagonal band (MS/DB), a key region for cognitive processing. Bilateral injection of αSyn fibrils into the dorsal striatum potently impaired cognition in mice. The cognitive decline was accompanied by accumulation of phosphorylated αSyn at Ser129 and reduction of gamma-aminobutyric acid (GABA)-ergic but not cholinergic neurons in the MS/DB. Since we have demonstrated that fatty acid-binding protein 3 (FABP3) is critical for αSyn neurotoxicity in nigral dopaminergic neurons, we investigated whether FABP3 also participates in αSyn pathology in the MS/DB and cognitive decline. FABP3 was highly expressed in GABAergic but rarely in cholinergic neurons in the MS/DB. Notably, *Fabp3* deletion antagonized the accumulation of phosphorylated αSyn, decrease in GABAergic neurons, and cognitive impairment caused by αSyn fibrils. Overall, the present study indicates that FABP3 mediates αSyn neurotoxicity in septal GABAergic neurons and the resultant cognitive impairment, and that FABP3 in this subpopulation could be a therapeutic target for dementia in synucleinopathies.

## 1. Introduction

Synucleinopathies, including Parkinson’s disease (PD) and dementia with Lewy bodies (DLB), are neuropathologically characterized by Lewy bodies (LBs) and Lewy neurites (LNs), which contain α-synuclein (αSyn) in abundance [1]. The most prominent symptom is Parkinsonism caused by deposits of misfolded αSyn and denervation in nigrostriatal dopaminergic neurons [2]. Another severe symptom is dementia, which has been documented to occur in more than half of PD patients in longitudinal prospective cohort studies over a decade [3,4]. Cognitive impairment in DLB has been reported to be correlated with LBs and LNs in the cerebral cortex and the hippocampus in some postmortem studies [5,6]. However, while DLB is distinguished from PD by the onset of dementia prior to or within 1 year after that of Parkinsonism [7], αSyn pathologies in these cortical and limbic regions are observed at a relatively later in Braak stages [8]. While cortical and hippocampal αSyn pathologies are crucial for cognitive decline, there is likely to be involvement of other regions with the αSyn burden in cognitive symptoms at earlier stages.

The medial septum/diagonal band (MS/DB) is one of the subcortical regions projecting glutamatergic, gamma-aminobutyric acid (GABA)-ergic, and cholinergic neurons to various areas, including the hippocampus and cerebral cortex [9,10,11]. These neurons in the MS/DB are crucial for the generation of neuronal oscillations known as theta rhythms (3–12 Hz), a critical process for cognition in particular, to regulate neural activity in limbic systems [12]. Among these subpopulations, septal glutamatergic neurons have been demonstrated to regulate firing of hippocampal CA1 neurons to initiate locomotion behaviors [13]. On the other hand, GABAergic and cholinergic neurons are largely involved in cognitive processing, including memory formation and spatial navigation via regulation of hippocampal long-term potentiation, synchronization of interneurons, and cortical neural activities [14,15,16,17,18,19]. The promotion of these behaviors is abolished by either selective or non-selective lesions of each neuronal subpopulation in the MS/DB [20,21,22,23], thereby supporting the significance of this region in multiple behaviors, including locomotion, memory, and sleep. Although αSyn pathology has been observed in the MS/DB at Braak stage 3, earlier than in the hippocampus and the cerebral cortex [8,24,25], detailed analyses in this region have not been conducted to date. Likewise, αSyn pathology in the MS/DB has been observed in animal models of intracerebral injection of αSyn fibrils, though the region has been presented as one of many comprehensive analyses and has not been focused on in terms of the mechanisms and cellular subpopulations [26,27,28]. The significance and molecular mechanisms of septal αSyn pathology in dementia should be elucidated.

Long-chain polyunsaturated fatty acids (LC-PUFAs) have been documented to be a factor responsible for αSyn aggregation and oligomerization. While LC-PUFAs, including docosahexaenoic acid and arachidonic acid, are essential for brain development [29], they could directly bind with and promote αSyn oligomerization in vitro using mesencephalic cultures and postmortem brains with synucleinopathies [30,31]. Fatty acid-binding proteins (FABPs) are a family of lipid chaperones that solubilize LC-PUFAs and traffic them to proper intracellular compartments [32]. In mammals, dopaminergic neurons in the substantia nigra pars compacta (SNpc) highly express FABP3 [33]. Notably, nigral FABP3 levels were increased in postmortem brains of PD patients compared to those of non-neurodegenerative controls [34]. We thus hypothesized that LC-PUFA induced αSyn cytotoxicity is mediated by FABP3 under physiological conditions and have investigated the same in serial studies. We found that FABP3 binds to αSyn and promotes its accumulation and oligomerization in dopaminergic neurons in the SNpc in 1-methyl-1,2,3,6-tetrahydropyridine (MPTP)-treated mice [33]. Moreover, FABP3 promoted αSyn phosphorylation and its spread through the brain in intranigral αSyn fibril-injected mice [35]. The αSyn cytotoxicity and the resultant Parkinsonism was antagonized in *Fabp3*^−/−^ mice, by treatment with a competitive inhibitor of LC-PUFAs binding to FABP3 [33,35,36,37]; however, it remains unclear whether FABP3 is also related to αSyn-induced cognitive impairment and septal αSyn pathology. Moreover, there has been no study investigating whether FABP3 is expressed in the MS/DB, and if so, in which subpopulation.

In the present study, we investigated the effects of αSyn strains on cognitive function, neuropathology in the MS/DB, and the relationship of FABP3. Intrastriatal injection of αSyn fibrils severely impaired cognition, and phosphorylated αSyn (pαSyn) at Ser129 accumulated in GABAergic but not in cholinergic neurons in the MS/DB. Notably, FABP3 was highly expressed in GABAergic neurons in the MS/DB, and its deletion antagonized αSyn fibril-induced cytotoxicity in GABAergic neurons and cognitive impairment.

## 2. Results

### 2.1. Distinct αSyn Strains Impair Cognition after Intrastriatal Injection in Mice

While αSyn fibrils and ribbons are reported to impair motor function after intracerebral injection [38], their effects on cognitive function remains unclear. To address this issue, we first injected αSyn strains into the dorsal striatum of mice and subjected them to behavioral analyses at 30 and 60 days after the injection (Experiment 1 in Figure 1A). In the novel object recognition task, mice receiving monomeric αSyn could discriminate novel objects from familiar counterparts (Figure 2A). In contrast, αSyn fibrils impaired object recognition memory 30 days after injection (Figure 2A). αSyn ribbons also impaired the discrimination ability of mice at 60 days (Figure 2A). In the passive avoidance task, there was a significant effect of αSyn strains on latency to enter the dark compartment at both 30 (F_(3,24)_ = 7.02, *p* = 0.002) and 60 days (F_(3,24)_ = 18.17, *p* < 0.001). Intrastriatal injection of αSyn fibrils significantly reduced the latency to enter the dark compartment (30 days: *p* < 0.01 vs. PBS-injected group; 60 days: *p* < 0.01 vs. PBS-injected group; Figure 2B). Intrastriatal injection of αSyn ribbons also impaired cognitive function in the passive avoidance task at 60 days (*p* < 0.01 vs. PBS-injected group; Figure 2B). When spatial working memory was assessed in the Y-maze task, there was no significant difference in terms of spontaneous alternation behaviors between groups (Figure 2C). To further evaluate spatial working memory in detail, the Barnes maze task was conducted (Figure 2D–G). In training sessions, spatial working memory was not affected by any αSyn strain at 30 days (Figure 2D). At 60 days, two-way repeated measures analysis of variance (ANOVA) detected significant differences in terms of the latency to enter the target hole by day (F_(2.37,56.78)_ = 24.12, *p* < 0.001), αSyn strain (F_(3,24)_ = 8.40, *p* < 0.001), and day × αSyn strain (F_(7.10,56.78)_ = 2.29, *p* = 0.039). On days 2 and 3 in the training sessions, αSyn fibrils significantly delayed the latency (day 2: *p* < 0.01 vs. PBS-injected group; day 3: *p* < 0.01 vs. PBS-injected group; Figure 2E). Four days after the final training sessions, a probe test was conducted (Figure 2F,G). At 30 days after the injection, all groups retained their spatial memory, although αSyn fibrils tended to impair the ability (Figure 2F). At 60 days, the latency to explore the target hole was significantly affected by the αSyn strains (F_(3,24)_ = 5.95, *p* = 0.003). αSyn fibrils significantly impaired the retention of spatial memory (*p* < 0.01 vs. PBS-injected group; Figure 2G). Taken together, intrastriatal injection of αSyn fibrils potently impaired cognitive function in mice.

### 2.2. αSyn Fibrils Reduce Septal GABAergic Neurons after Intrastriatal Injection in Mice

We then addressed which histological alterations contributed to the cognitive impairments seen in αSyn fibril-injected mice. Cholinergic neurons in the MS/DB are decreased in Alzheimer’s disease patients and model mice, and some DLB patients [39,40,41,42]. Thus, we quantified choline acetyltransferase (ChAT)-positive cells in the MS/DB and addressed whether these neurons are affected by αSyn fibril injection. Unexpectedly, the number of cholinergic neurons was not affected by any αSyn strain (Figure 3A). In contrast, αSyn strains had a significant effect on the number of glutamic acid decarboxylase 67 (GAD67)-positive GABAergic cells in the MS/DB (F_(3,12)_ = 7.66, *p* = 0.004). Notably, αSyn fibrils significantly reduced the number of GAD67-positive cells (*p* < 0.05 vs. PBS-injected group; Figure 3B).

### 2.3. αSyn Fibrils Induce pαSyn Accumulation in Septal GABAergic Neurons after Intrastriatal Injection in Mice

Next, we evaluated whether αSyn cytotoxicity is related to the reduction of GABAergic neurons in the MS/DB. To address this issue, pαSyn-positive cells in ChAT- or GAD67-positive cells were quantified. There was no significant difference between any αSyn strains (Figure 4A). In contrast, there was a significant effect of αSyn strains in terms of the percentage of pαSyn-positive cells/GAD67-positive cells (F_(2,9)_ = 15.17, *p* = 0.001). Consistent with the reduction of GAD67-positive cells (Figure 4B), αSyn fibrils significantly increased the number of pαSyn-positive cells related to GAD67-positive cells in the MS/DB (*p* < 0.01 vs. αSyn monomer-injected group; Figure 4B). ATTO550-positive reactivities in the MS/DB seemed to be similar between αSyn strains (Figure 4A,B). pαSyn accumulation in GABAergic neurons was further confirmed by the accumulation in parvalbumin-positive cells (Appendix A).

### 2.4. Properties of Expression and Localization of αSyn and FABP3 in MS/DB

Previous studies suggest that the extent of αSyn spreading and its cytotoxicity partly depend on neural projection and expression levels of endogenous αSyn [43,44,45]. We recently analyzed temporal profiles of αSyn propagation in the SNpc after intrastriatal injection of αSyn monomers and fibrils and demonstrated αSyn strain-dependent patterns of internalization in dopaminergic neurons in the SNpc, which potently innervates the dorsal striatum [37]. Unlike the SNpc, MS/DB has only slight projections with the dorsal striatum for both antero- and retrogradely [46,47]. Consistent with the extent of neural projection, intraneuronal accumulation of exogenous αSyn strains in the MS/DB was slight and sparse over the time course, although αSyn fibrils tended to accumulate slowly and progressively (Appendix A), in line with those observed in the SNpc [37].

Expression levels of endogenous αSyn in the MS/DB should also be investigated as another factor that contributes to αSyn spreading and the toxicity. Additionally, it remains unclear whether FABP3, a critical molecule for αSyn neurotoxicity in the SNpc [33,35,37], is expressed in the MS/DB. To clarify the expression levels of these proteins in the MS/DB, lysates from naïve mice were subjected to immunoblotting analyses and the levels were compared to those in the SNpc (Figure 5A). When endogenous αSyn levels were quantified, there was no significant region effect (Figure 5B). In contrast, the expression levels of FABP3 were significantly higher in the MS/DB than in the SNpc (t_(7.40)_ = 3.58, *p* < 0.01; Figure 5B). To further clarify the detailed localization of αSyn and FABP3 in the MS/DB, the neuronal distribution of each protein was investigated (Figure 5C–E). Similar to the SNpc and other brain regions, αSyn was abundant in presynaptic terminals, occasionally positive for cytoplasm (Figure 5C). Both ChAT- and GAD67-positive cells did not contain rich αSyn (Figure 5C). As reported that FABP3 is expressed in mature neurons [48], FABP3 is localized in microtubule-associated protein 2 (MAP2)-positive neurons in the MS/DB (Figure 5D). Notably, FABP3 was localized rarely in ChAT-positive but highly in GAD67-positive cells (Figure 5D,E). Collectively, while αSyn is present in the MS/DB with typical localization similar to other brain regions, FABP3 is abundantly and preferentially expressed in GABAergic neurons in this region. Moreover, pαSyn accumulation was observed in FABP3-positive cells in the MS/DB of αSyn fibril-injected mice, supporting the significance of FABP3 expression in αSyn cytotoxicity in GABAergic neurons in the MS/DB (Appendix A).

### 2.5. Fabp3 Deletion Inhibits pαSyn Accumulation and Reduction of GABAergic Neurons in MS/DB after Intrastriatal Injection of αSyn Fibrils in Mice

To address whether FABP3 contributes to αSyn fibril-induced cytotoxicity in the MS/DB, we injected αSyn fibrils in *Fabp3*^−/−^ mice and evaluated pαSyn accumulation and decrease in GABAergic neurons. pαSyn accumulation in GAD67-positive cells was significantly reduced in *Fabp3*^−/−^ mice compared to *Fabp3*^+/+^ counterparts (t_(7.35)_ = 5.12, *p* < 0.01; Figure 6A,B). Additionally, in terms of GAD67-positive cells, two-way ANOVA detected significant effects of genotype (F_(1,20)_ = 14.27, *p* = 0.001), αSyn strain (F_(1,20)_ = 5.78, *p* = 0.026), and genotype × αSyn strain (F_(1,20)_ = 7.51, *p* = 0.013). Decreased GAD67-positive cells seen in the MS/DB of αSyn fibril-injected *Fabp3*^+/+^ mice were restored in *Fabp3*^−/−^ counterparts (*p* < 0.01; Figure 6C,D). These results suggest that *Fabp3* deletion attenuated αSyn neurotoxicity and the resultant decrease in septal GABAergic neurons.

### 2.6. Fabp3 Deletion Restores αSyn Fibril-Induced Cognitive Impairments after Intrastriatal Injection in Mice

Finally, we investigated the effects of *Fabp3* deletion on cognitive impairments caused by intrastriatal injection of αSyn fibrils. Since *Fabp3*^−/−^ mice display deficits in exploration behaviors in the novel object recognition task [49,50], cognitive function was assessed with the passive avoidance and Barnes maze tasks. In terms of the latency to enter the dark compartment in the passive avoidance task, there were significant effects of genotype (30 days: F_(1,36)_ = 6.48, *p* = 0.015; 60 days: F_(1,36)_ = 4.25, *p* = 0.047), αSyn strain (30 days: F_(2,36)_ = 11.16, *p* < 0.001; 60 days: F_(2,36)_ = 9.21, *p* < 0.001), and genotype × αSyn strain (30 days: F_(2,36)_ = 12.93, *p* < 0.001; 60 days: F_(2,36)_ = 10.58, *p* < 0.001). The shortened retention time seen in αSyn fibril-injected *Fabp3*^+/+^ mice was potently restored in *Fabp3*^−/−^ counterparts (30 days: *p* < 0.01; 60 days: *p* < 0.01; Figure 7A). In the Barnes maze task, a three-way repeated measures ANOVA detected significant differences in terms of the latency to enter the target hole by day (F_(2.41,86.72)_ = 16.47, *p* < 0.001), genotype (F_(1,36)_ = 17.91, *p* < 0.001), αSyn strain (F_(2,36)_ = 7.73, *p* = 0.002), day × genotype (F_(2.41,86.72)_ = 3.01, *p* = 0.045), day × αSyn strain (F_(4.82,86.72)_ = 3.51, *p* = 0.007), genotype × αSyn strain (F_(2,36)_ = 8.85, *p* < 0.001), day × genotype × αSyn strain (F_(4.82,86.72)_ = 2.50, *p* = 0.039). Consistent with the results above (Figure 2E), αSyn fibrils impaired spatial learning in *Fabp3*^+/+^ mice (day 2: *p* < 0.01 vs. PBS-injected *Fabp3*^+/+^ group; day 3: *p* < 0.01 vs. PBS-injected *Fabp3*^+/+^ group; Figure 7B,C). These impairments were significantly attenuated in *Fabp3*^−/−^ counterparts (day 2: *p* < 0.01; day 3: *p* < 0.01; Figure 7B,C). In the probe test, there were significant effects of the αSyn strain (F_(2,36)_ = 5.16, *p* = 0.011) and genotype × αSyn strain (F_(2,36)_ = 4.79, *p* = 0.014) in terms of the latency to explore the target hole. *Fabp3* deletion significantly improved the impaired retention of spatial memory seen in αSyn fibril-injected mice (*p* < 0.05; Figure 7D).

## 3. Discussion

In the present study, we documented the following findings: (1) Intrastriatal injection of αSyn fibrils leads to reduction of GABAergic neurons in the MS/DB and cognitive impairment. (2) FABP3 is highly expressed in GABAergic neurons in the MS/DB. (3) *Fabp3* deletion suppresses pαSyn accumulation, reduction in GABAergic neurons, and resultant cognitive impairments. Thus, FABP3 may partly contribute to cognitive impairments due to septal GABAergic lesions in synucleinopathies.

We previously demonstrated that while FABP3 and αSyn make complexes and oligomerize in the SNpc of MPTP-treated mice, overexpression of the proteins fails to promote αSyn oligomerization in the absence of 1-methyl-4-phenylpyridinium (MPP^+^) in PC12 cells [33]. Arachidonic acid also failed to promote MPP^+^-induced αSyn oligomerization and the resultant cell death when mutated FABP3 (F16S) that has little affinity for LC-PUFA binding is transfected in PC12 cells [33]. Notably, binding affinities for arachidonic acid of FABP3 is 10–100 times compared to those of αSyn [51,52,53], suggesting that LC-PUFAs preferentially bind to FABP3 rather than αSyn under physiological condition. Furthermore, our recent studies demonstrated that a FABP3-selective ligand that competitively inhibits LC-PUFA binding attenuated αSyn oligomerization in MPTP-treated mice and pαSyn accumulation in mice with intranigral injection of αSyn fibrils [35,36]. Overall, there reports suggest that LC-PUFA-bound FABP3 makes complexes with αSyn and thereafter promotes its oligomerization and aggregation under pathological conditions such as oxidative stress in synucleinopathies.

The extent of αSyn propagation and neurotoxicity have been shown to partly depend on conformational differences. A recent study demonstrated that αSyn polymorphs in brain lysates of DLB patients were different from those of PD and multiple system atrophy (MSA), and intranigral injection of these patient-derived αSyn in mice induced nigrostriatal dopaminergic degeneration and motor impairments with relative severity in patients with MSA [54]. Another study reported that αSyn polymorphs in the brains of patients with PD were distinct from those of MSA patients [55]. These polymorph-dependent phenotypes could also be applied to synthetic αSyn strains; αSyn fibrils exerted cytotoxicity in cultured cells, and decreased striatal dopaminergic fibers and impaired motor function in intranigral injected rats more potently than ribbon species [38,56]. Rey et al. also demonstrated strain-dependent αSyn propagation and localization of aggregated forms [57]. In the present study, cognitive impairments were seen in mice receiving fibrils followed by ribbons at a relatively earlier time after intrastriatal injection. Consistent with these results, pαSyn accumulation and reduction of GABAergic neurons in the MS/DB were evident in αSyn fibril-injected mice, supporting the hypothesis that αSyn properties regarding propagation and cytotoxicity are partly derived from polymorphs of each strain. Further progress is expected in the study regarding the relationship between distinct αSyn strains and phenotypes in synucleinopathies.

As cholinergic neuronal loss in the MS/DB and the decreased levels in the cerebral cortex have been reported in DLB patients [40,58], a cholinesterase inhibitor donepezil is used to ameliorate cognitive decline in DLB. A series of randomized clinical trials have shown that donepezil was mostly effective for the cognitive impairment seen in DLB patients [59,60,61,62]. In the present study, intrastriatal injection of αSyn fibrils reduced the cell number of GABAergic neurons without changes in that of cholinergic subpopulations in the MS/DB. GABAergic neurons in the MS/DB are crucial for learning and memory processing. Among septal neurons projecting to the hippocampal CA1 region, more than 80% are GABAergic, and inhibition or lesions of these subpopulations impair contextual memory and spatial navigation [19,63,64]. Selective activation of septohippocampal GABAergic neurons potentiates hippocampal theta oscillations and object exploration in mice [65]. Since septal GABAergic neurons project to the cell bodies and dendrites of parvalbumin or calbindin-positive interneurons in the hippocampus [66], the reduction of septal GABAergic neurons would lead to hyperexcitability of these interneurons and impairments in synchronization of neural networks in the hippocampus. Hyperexcitability of parvalbumin-positive interneurons in the hippocampus enhances the vulnerability to amyloid-beta, disrupts synaptic transmission, and impairs spatial memory in mice [67]. Dysfunction of parvalbumin-positive interneurons and gamma oscillations in the hippocampus are also suggested to be related to neuropathology in DLB patients [68]. Gamma oscillations in the hippocampus and sleep-related ones in the cortex were altered prior to cognitive decline in mice carrying human mutated *Snca* (A30P) compared to wild-type counterparts [69,70,71]. These reports support that disruption of GABAergic in addition to cholinergic neurons in the MS/DB would contribute to impaired neural activity in the hippocampus and cortex, and cognitive impairment in synucleinopathies.

Endogenous αSyn in naïve mice represented typical patterns of localization in the MS/DB, similar to other regions, as dense in presynaptic terminals and occasionally positive in cytoplasm. In addition, αSyn was not concentrated in either cholinergic or GABAergic neurons in the MS/DB. The spread of αSyn pathology was not observed in *Snca*-deficient mice injected with αSyn fibrils [72], supporting the hypothesis that the extent of αSyn spreading through the brain depends on the expression levels of endogenous αSyn in addition to neural projection [43,44,45]. Combined analyses using intracerebral injection and a mathematical model regarding brain connectome revealed that αSyn propagates mainly in a retrograde, followed by an anterograde manner over the experimental time course [73]. Septal αSyn pathology was evident after overexpression of human mutated *Snca* (A53T) in the locus coeruleus, which has projections with the septal nucleus [74,75]. Injection of αSyn fibrils in the entorhinal cortex, which also has projections with the septal nucleus, induced septal αSyn pathology more severely than intranigral injection [72,76,77]. These results further support the relationship between αSyn propagation and neural projections. In the present study, αSyn was injected into the dorsal striatum. The MS/DB has few projections both anterogradely and retrogradely with the dorsal striatum [46,47]. Nevertheless, accumulation of exogenously applied αSyn and pαSyn was observed in the MS/DB, and the latter was particularly evident in GABAergic neurons. We have documented that FABP3 binds with αSyn to promote its oligomerization and propagation, and FABP3 inhibition by a selective ligand or gene deletion potently suppresses these αSyn pathologies, especially in dopaminergic neurons in the SNpc [33,35,36,37]. Since FABP3 is highly expressed in these neurons [33], we have proposed the protein as a potential enhancer of αSyn pathology and Parkinsonism. It is worth noting that FABP3 inhibition and deletion restored cognitive impairments seen in mice with intranigral injection of αSyn [35]; however, it remains unclear whether the contribution of FABP3 is restricted within dopaminergic neurons in the SNpc and how the protein is related to αSyn-induced cognitive impairments. Here, we found that FABP3 is highly expressed in the MS/DB compared to the SNpc and is abundant in GABAergic but sparce in cholinergic neurons, unlike endogenous αSyn. Since the histological features and cognitive impairments seen in αSyn fibril-injected mice were suppressed by *Fabp3* deletion, it is likely that FABP3 participates in αSyn pathology in not only nigral dopaminergic neurons but also other regions whose cell subpopulations highly express FABP3, such as septal GABAergic neurons. Further studies using region- or cellular subpopulation-specific knockdown of *Fabp3* gene would warrant our hypothesis that the vulnerability to αSyn pathology in distinct cellular subpopulations is attributed to the expression levels of FABP3, in addition to those of endogenous αSyn and the extent of neural projection. In addition, αSyn levels of higher molecular weight and insoluble species should also be investigated in the MS/DB in this model and the relevance of FABP3 in order to further highlight the importance of αSyn pathology in the region for cognitive decline.

In conclusion, the present study documented that FABP3 partly contributes to αSyn propagation in GABAergic neurons in the MS/DB and cognitive impairments seen in mice injected intrastriatally with αSyn fibrils. αSyn pathology was well correlated with distinct localization of FABP3 in the MS/DB, and *Fabp3* deletion potently inhibited the pathology and cognitive impairments. These results highlighted the importance of FABP3 expression in αSyn pathology. Taken together, FABP3 could be a molecular target for both Parkinsonism and dementia in synucleinopathies.

## 4. Materials and Methods

### 4.1. Animals and Stereotaxic Surgery

All animal experiments were approved by the Committee on Animal Experiments at Tohoku University (2017PhA-001; 2020PhA-007) and conformed to the Regulations for Animal Experiments and Related Activities at Tohoku University. We made every effort to minimize the number of mice used and their discomfort. Wild-type C57BL/6 (*Fabp3*^+/+^) male mice (Japan SLC, Shizuoka, Japan) were bred in a conventional environment with a 12/12-h light/dark cycle (light: 9:00–21:00) and freely accessible normal chow and water. *Fabp3*^−/−^ male mice were generated on a C57BL/6 background [78], and gender- and age-matched mice were used.

Purification and ATTO550-labelling of αSyn assemblies termed monomer, fibril, and ribbon were performed as previously described [38,56]. At the age of three months, mice were subjected to stereotaxic surgery. Under anesthesia, mice received these materials (5 μg/μL in sterile PBS) or PBS with a volume of 1 μL per hemisphere in the bilateral dorsal striatum (coordinates: AP: +0.8 mm, ML: 2.0 mm, DV: 2.6 mm; according to [79]).

### 4.2. Behavioral Analyses

Behavioral tasks were performed at 30 and 60 days after stereotaxic injection. Mice were separately prepared for each time point in order to exclude the effects of memory retention due to repeated exposure to the same task. During all tasks, the instruments were cleaned with 70% ethanol after each session to eliminate odor cues.

#### 4.2.1. Novel Object Recognition Task

The novel object recognition task was performed as previously described [36]. In trial sessions, mice were subjected to a home-cage containing two identical cubic objects at symmetric positions for 10 min. One hour after the session, one of the objects was replaced with a different-shaped object (novel), and exploratory behaviors were recorded for 5 min. These behaviors were defined as touching, rearing on, and sniffing from distances less than 10 mm. Object recognition was calculated as exploration of a novel object/both objects (%) and represented as a discrimination index.

#### 4.2.2. Step-Through Passive Avoidance Task

The step-through passive avoidance task was performed as previously described [36]. The apparatus consisted of dark (14 × 17.5 × 15 cm) and light (9 × 11.5 × 15 cm) compartments with a rod floor made of stainless steel (STC-001M; Muromachi Kikai, Tokyo, Japan). A manual electric stimulus-generating scrambler (SGS-003DX; Muromachi Kikai) was connected to the rods in the dark box. Mice were placed in the light box for 60 s, and then the partition between the compartments was opened to allow entrance into the dark compartment. On entering, the partition was immediately closed and mice were subjected to an electric stimulus (0.6 mA for 3 s). Thirty seconds later, mice were pulled up from the apparatus and allowed to return to their home-cages. The next day, mice were placed in the light box again with the partition opening, and the step-through latency to enter the dark compartment was recorded for up to 300 s.

#### 4.2.3. Y-Maze Task

Short-term spatial reference memory was assessed using a Y-maze task as previously described [80]. Mice were first placed at the end of one arm of the maze and allowed to explore freely for 8 min. A correct alternation was defined as sequential entries into all three arms. The maximum number of alternations was defined as the number of total arm entries − 2, and the percentage of alternations was calculated as actual/maximal alternations × 100.

#### 4.2.4. Barnes Maze Task

The Barnes maze task was performed to assess long-term spatial reference memory as previously described [81]. The apparatus was made up of a circular platform (diameter: 90 cm) with equally spaced 12 holes (diameter: 5 cm) along the perimeter at a distance of 1.8 cm from the edge. Only one (target hole) led to an escape box (13 × 35 × 8 cm) below the platform. During the task, the target hole was located at the same position. Bright light was located 1.5 m above the platform as an aversive stimulus to encourage exploration. During the task, the observer and the surrounding materials were located at the same position as the landmarks. The task consisted of 4-day training sessions and 1-day probe test sessions. Only on the first day of training sessions, mice were first subjected to a habituation phase. They were placed in the center of the platform and allowed to explore the maze freely for up to 1 min. In cases where 1 min had elapsed before entering, mice were covered with a clear beaker and gently navigated and allowed to enter, and then allowed to stay in the box for 1 min. After 1 h, they were subjected to training sessions. At the beginning of the session, mice were placed inside an opaque cylinder (diameter, 7 cm; height, 16 cm), which allowed them to face a random direction. After 15 s, the cylinder was lifted up and a buzzer was immediately turned on to further encourage the exploration behaviors. The session was finished when mice entered the escape box or 2 min had elapsed. During the latter, mice were navigated to the escape box in the same manner as the habituation phase. After approximately 2 h, they were subjected to the session again. Four days after the final training sessions (day 8), the mice were subjected to probe test sessions. They were allowed to explore the maze in the same manner as the training sessions, but without the escape box. In probe test sessions, the cumulative latency to explore the target hole was recorded. Exploration behaviors were defined as nose pokes and head deflections.

### 4.3. Immunofluorescence and Quantification

Each mouse was perfused and fixed with 4% paraformaldehyde solution, and the brain was post-fixed for 18–36 h at 4 °C. For the localization analyses of endogenous αSyn and FABP3, naïve *Fabp3*^+/+^ male mice received the same procedures at the age of three months. The brain was cut into 50 μm coronal sections using a vibratome (DTK-1000; Dosaka EM, Kyoto, Japan). Immunofluorescence was conducted as previously described [36]. Sections were blocked with 5% normal donkey serum in PBS for 30 min at room temperature and incubated with primary antibodies overnight at 4 °C. These were then probed with secondary antibodies for 60–90 min at room temperature and coverslipped with a mounting medium (RRID: AB_2336789; Vector Laboratories, Burlingame, CA, USA). Some sections were counterstained with 4′,6-diamidino-2-phenylindole. Immunoreaction of FABP3 was visualized by tyramide signal amplification according to the manufacturer’s instructions (PerkinElmer, Waltham, MA, USA) as previously described [33]. Primary and secondary antibodies are summarized in Appendix A.

Images were acquired by a confocal microscope TCS SP8 (Leica Microsystems, Wetzlar, Germany) and the z-stacks were reconstructed to 3D images with a software LAS X (RRID: SCR_013673; Leica Microsystems). The images of ChAT- or GAD67-positive cells in the entire MS/DB were captured with 10× or 20× objective lens, followed by tile-stitch processing using LAS X. The acquired images (three sections per mouse) were analyzed and counted in Fiji image J (Ver. 1.52; RRID: SCR_003070; National Institutes of Health, Bethesda, MD, USA). For quantification of pαSyn, the images were captured with a 63× objective lens (six images from three sections per mouse). The anteroposterior coordinates from the bregma were 1.18 to 0.74 mm as described (Paxinos and Franklin, 2001).

### 4.4. Immunoblotting Analyses

Protein extraction and immunoblotting analyses were conducted as previously described [36]. MS/DB and SNpc regions were separated from naïve *Fabp3*^+/+^ and *Fabp3*^−/−^ male mice at the age of three months. Tissues were lysed in homogenizing buffer (500 mM NaCl, 50 mM Tris-HCl, pH 7.5) with protease and phosphatase inhibitors and were centrifuged (15,000 rpm, 4 °C, 10 min). The resultant supernatant was mixed with Laemmli’s buffer and boiled for 5 min. Proteins were separated in 13.5% SDS-PAGE and blotted on polyvinylidene difluoride membranes. Blots for αSyn were then crosslinked with 4% paraformaldehyde and 0.1% glutaraldehyde for 30 min at room temperature [82]. After blocking with 3% BSA in Tris-buffered saline with 0.1% Tween 20, membranes were probed with primary antibodies overnight at 4 °C, and then probed with secondary antibodies for 60 min at room temperature. Blots were visualized using an ECL detection system with ImageQuant LAS-4000 mini (GE Healthcare, Chicago, IL, USA). Primary and secondary antibodies are summarized in Appendix A.

### 4.5. Statistical Analyses

The results are presented as box plots with whiskers delineating the 95th and 5th percentiles. The first and third quartiles are depicted by the bottom and top of the box, respectively. The median is represented by the line within the box. Statistical analyses were performed using JASP (ver. 0.13.1; RRID: SCR_015823; University of Amsterdam, Amsterdam, The Netherlands). Statistical significance was detected using two-tailed paired *t*-test for the results of the novel object recognition task (Figure 2A) or two-tailed Welch’s *t*-test for the other comparisons between the two groups. For multiple-group comparisons, the significance was detected by three-way (genotype × strain × time), two-way (genotype × time or genotype × strain), or one-way ANOVA followed by post-hoc Tukey’s test. When repeated measures ANOVA was performed (for training sessions in the Barnes maze task; Figure 2D,E and Figure 7B,C), homogeneity of variance within subjects was checked by Mauchly’s sphericity test, and if *p* values were <0.05 (i.e., the sphericity was violated), followed by detection of F values by Greenhouse-Geisser ε correction. When ε values (estimating sphericity) > 0.1, corrected F values were adopted. Values of significant difference between groups were set as follows: * when *p* < 0.05, and ** when *p* < 0.01.

## Figures and Tables

**Figure 1 ijms-22-00400-f001:**
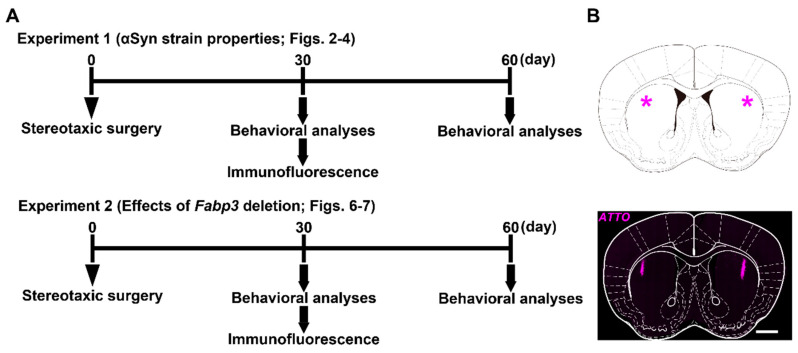
Experimental schedule and the injection site in mouse brain. (**A**) Experimental schedule in this study using intrastriatally α-synuclein (αSyn)-injected mice. Mice were divided into two groups, received the stereotaxic surgery, and were subjected to either Experiment 1 or 2. Experiment 1 was conducted to reveal αSyn strain-dependent properties regarding cognitive impairment and the related pathological alterations. Mice were injected with αSyn monomers, fibrils, or ribbons. Experiment 2 was conducted using fatty acid-binding protein 3 (*Fabp3)*^−/−^ mice to address whether FABP3 participates in αSyn-induced cognitive impairment and the related pathology. Mice were injected with αSyn monomers or fibrils. (**B**) (Top) The injection site (*) is depicted. (Bottom) The actual site confirmed by direct visualization of immunoreactivity against ATTO550 (magenta). The image was obtained 30 days after the injection of αSyn fibrils and was overlaid on the diagram shown above. Scale bar: 1 mm.

**Figure 2 ijms-22-00400-f002:**
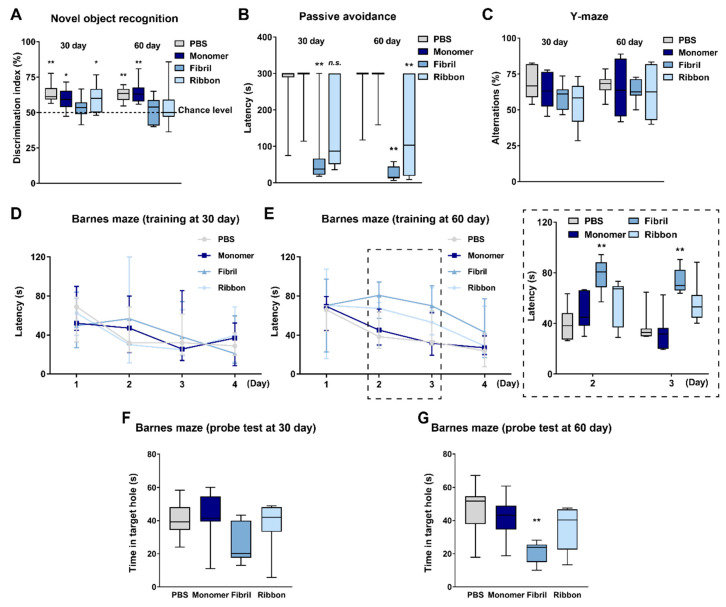
αSyn strain-dependent cognitive impairments in intrastriatally injected mice. (**A**) Discrimination index in a novel object recognition task (*n* = 7 per group). * *p* < 0.05 and ** *p* < 0.01 between contacts with familiar and novel objects within groups. (**B**) Latency to enter the dark compartment in a passive avoidance task (*n* = 7 per group). ** *p* < 0.01 vs. PBS-injected group in the same day. n.s., not significant. (**C**) Spontaneous alternations in a Y-maze task (*n* = 7 per group). (**D**,**E**) Latency to enter the escape box in training sessions in a Barnes maze task (*n* = 7 per group). The sessions were conducted twice per day. The insertion depicts the days when the latency was significantly different between groups. ** *p* < 0.01 vs. PBS-injected group. (**F**,**G**) Latency to explore the target hole in probe test sessions in the Barnes maze task (*n* = 7 per group). The test was conducted 7 days after the first training sessions. ** *p* < 0.01 vs. PBS-injected group.

**Figure 3 ijms-22-00400-f003:**
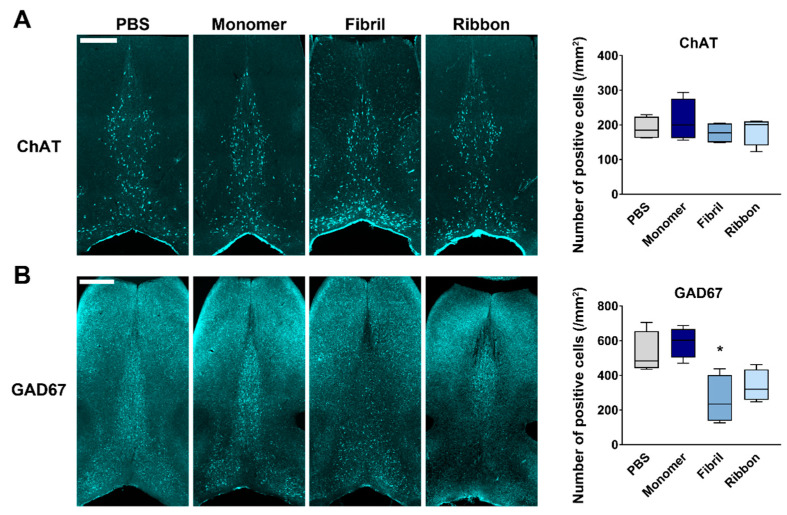
Selective decrease in septal GABAergic neurons after intrastriatal injection of αSyn. (**Left**) Representative images of choline acetyltransferase (ChAT)-positive (**A**) and glutamic acid decarboxylase 67 (GAD67)-positive (**B**) cells in the medial septum/diagonal band (MS/DB), 30 days after the injection. Scale bar: 500 μm. (**Right**) Quantitative analyses of each protein-positive cell number in the MS/DB. * *p* < 0.05 vs. PBS-injected group (*n* = 4 per group).

**Figure 4 ijms-22-00400-f004:**
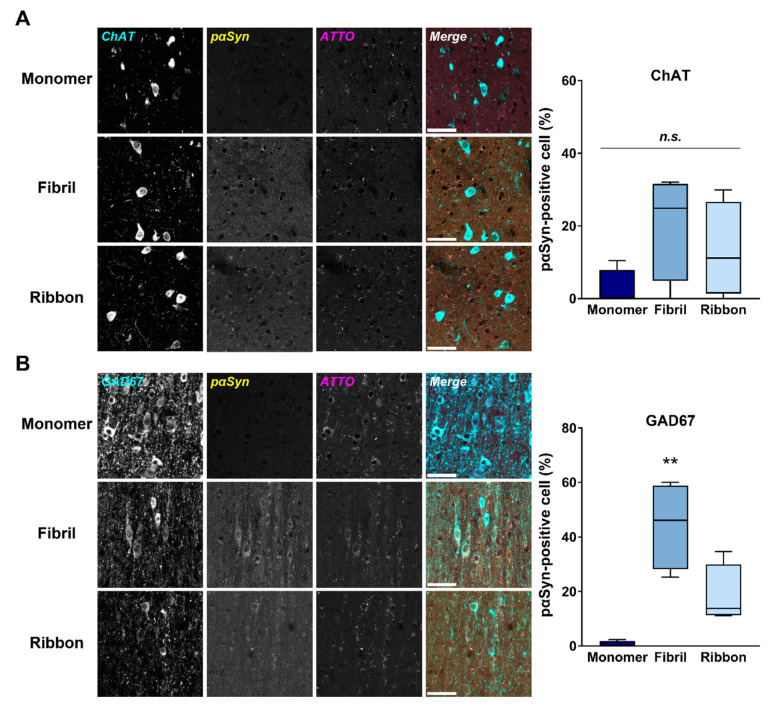
Accumulation of pαSyn in septal GABAergic neurons after intrastriatal injection of αSyn. (**Left**) Representative images of phosphorylated αSyn (pαSyn; yellow) accumulation and exogenously applied αSyn (ATTO; magenta) in ChAT-positive (**A**) and GAD67-positive (**B**) cells (cyan) in the MS/DB, 30 days after the injection. Scale bar: 50 μm. (**Right**) Quantitative analyses of the ratio of pαSyn-positive cells in ChAT-positive (**A**) and GAD67-positive cells. ** *p* < 0.01 vs. monomeric αSyn-injected group (*n* = 4 per group).

**Figure 5 ijms-22-00400-f005:**
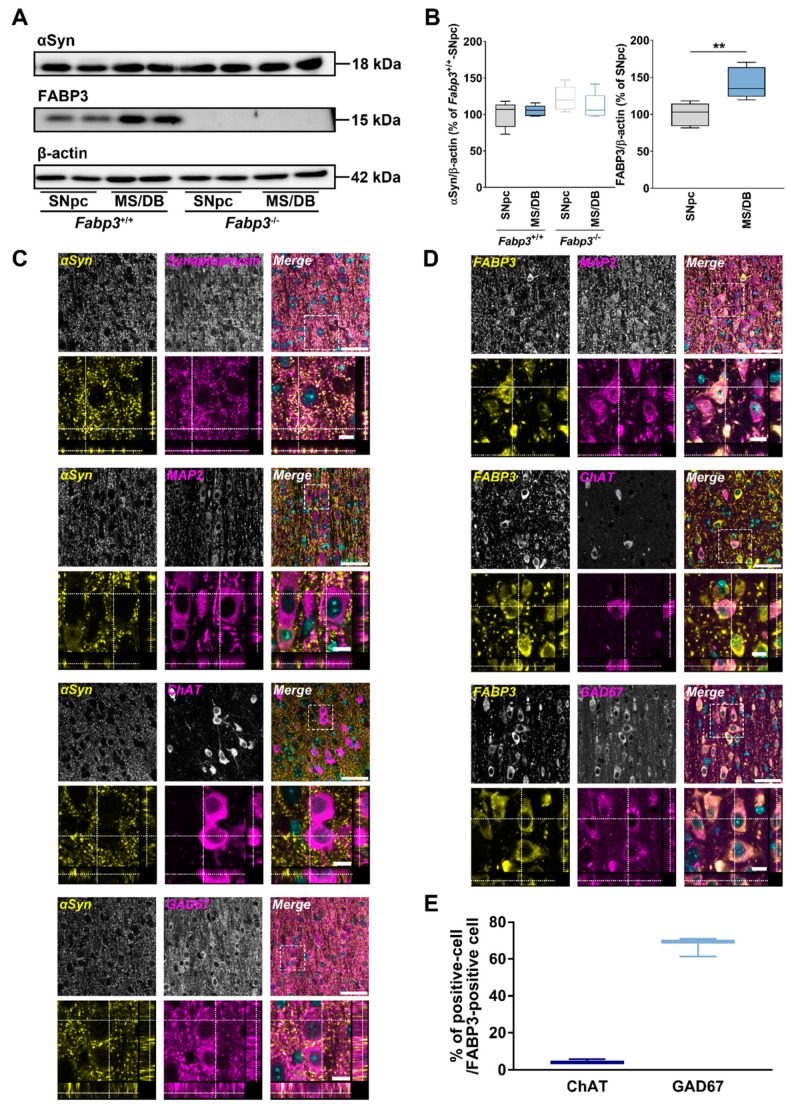
Properties of expression levels and the localization of endogenous αSyn and FABP3 in the MS/DB. (**A**) Representative images of blots probed with anti-αSyn and FABP3 antibodies detected in lysates of substantia nigra pars compacta (SNpc) and MS/DB from naïve *Fabp3*^+/+^ and *Fabp3*^−/−^ mice. (**B**) Quantitative analyses of the blots. ** *p* < 0.01 between SNpc and MS/DB from *Fabp3*^+/+^ mice (*n* = 5 per group). (**C**,**D**) Representative images of the localization of endogenous αSyn (**C**) and FABP3 (**D**) in the MS/DB. Sections were probed with anti-αSyn (**C**) or anti-FABP3 (**D**) antibody (yellow) with anti-synaptophysin, microtubule-associated protein 2 (MAP2), ChAT, or GAD67 antibodies (magenta). The lower images of each staining represent the magnification and the orthogonal XZ and YZ planes. Scale bar: 50 (merge) and 10 (magnification) μm, respectively. Sections were counterstained with 4′,6-diamidino-2-phenylindole (cyan). (**E**) Quantitative analyses of the ratio of ChAT-positive or GAD67-positive cells in FABP3-positive cells in the MS/DB (*n* = 3 each).

**Figure 6 ijms-22-00400-f006:**
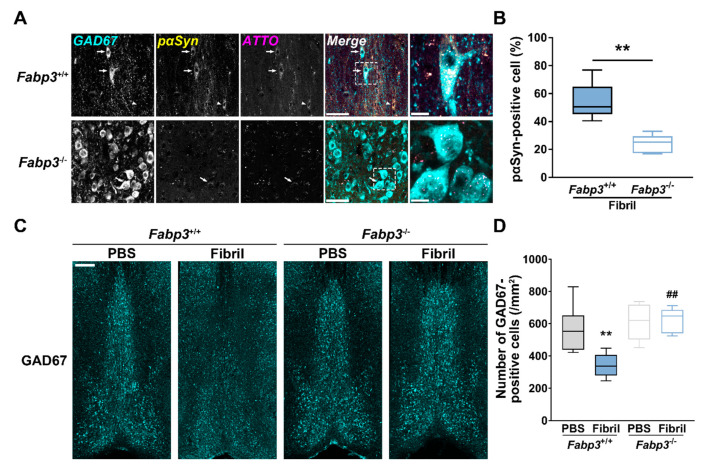
Effects of *Fabp3* deletion on pαSyn accumulation and reduction of septal GABAergic neurons after intrastriatal αSyn injection. (**A**) Representative images of pαSyn (yellow) accumulation in GAD67-positive cells (cyan) in the MS/DB, 30 days after the injection. Arrows and arrowheads depict pαSyn-accumulated GAD67-positive and -negative cells, respectively. Scale bar: 50 (merge) and 10 (magnification) μm, respectively. (**B**) Quantitative analyses of the ratio of pαSyn-positive cells in GAD67-positive cells (*n* = 6 per group). ** *p* < 0.01 between αSyn fibril-injected *Fabp3*^+/+^ and *Fabp3*^−/−^ mice. (**C**) Representative images of GAD67-positive cells (cyan) in the MS/DB, 30 days after the injection. Scale bar: 250 μm. (**D**) Quantitative analyses of GAD67-positive cells (*n* = 6 per group). ** *p* < 0.01 vs. PBS-injected *Fabp3*^+/+^ mice; ^##^
*p* < 0.01 vs. αSyn fibril-injected *Fabp3*^+/+^ mice.

**Figure 7 ijms-22-00400-f007:**
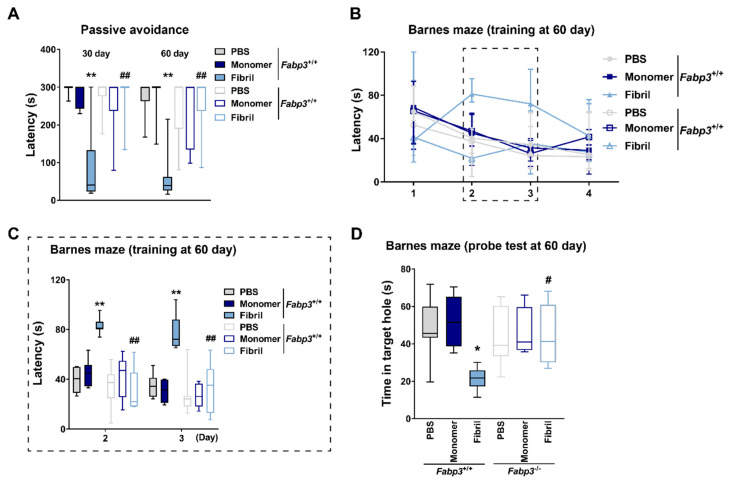
Effects of *Fabp3* deletion on cognitive impairments after intrastriatal αSyn injection. (**A**) Latency to enter the dark box in the passive avoidance task (*n* = 7 per group). ** *p* < 0.01 vs. PBS-injected *Fabp3*^+/+^ mice and ^##^
*p* < 0.01 vs. αSyn fibril-injected *Fabp3*^+/+^ mice in the same day. (**B**,**C**) Latency to enter the escape box in training sessions in the Barnes maze task (*n* = 7 per group). The sessions were conducted twice per day. The insertion (**C**) depicts the days when the latency was significantly different between groups. ** *p* < 0.01 vs. PBS-injected *Fabp3*^+/+^ mice; ^##^
*p* < 0.01 vs. αSyn fibril-injected *Fabp3*^+/+^ mice. (**D**) Latency to explore the target hole in the probe test sessions in the Barnes maze task (*n* = 7 per group). The test was conducted 7 days after the first training sessions. * *p* < 0.05 vs. PBS-injected *Fabp3*^+/+^ mice. ^#^
*p* < 0.05 vs. αSyn fibril-injected *Fabp3*^+/+^ mice.

## Data Availability

The data that support the findings of this study are available from the corresponding author upon reasonable request.

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
