# Peer review of "Crucial Role of FABP3 in αSyn-Induced Reduction of Septal GABAergic Neurons and Cognitive Decline in Mice"

_ijms, 2021, doi:10.3390/ijms22010400_

Round 1

Reviewer 1 Report

The authors investigated the effects of αSyn strains on cognitive function,  neuropathology in the MS/nDB, and the relationship of FABP3. Intrastriatal injection of αSyn fibrils  severely impaired cognition, and phosphorylated αSyn (pαSyn) at Ser129 accumulated in  GABAergic but not in cholinergic neurons in the MS/nDB. A key result of the study is that, FABP3 was highly expressed in GABAergic neurons in the MS/nDB, and its deletion antagonized αSyn fibril-induced cytotoxicity in GABAergic neurons and cognitive impairments. 

The cognitive decline in Parkinson's disease and other synucleinopathies is still not well studied and not well managed in clinical practice. Thus, the reasearch is of high potential interest.

The study is clean and well conducted and the manuscript is well-written.

Thus, my specific comments are only ir order to help with the publication, which I find highly reccomended.

nDB: Please use the standard abbreviations. The diagonal band is usually abreviated as DB or DBB (Diagonal band of Broca): Medial septum/diagonal band (MS/DB)

Introduction: 50-58: It should be mentioned that the MS/DB is particularly involved in the generation of theta rhythm in the hippocampus, which is key for cognitive processing. That is well discussed further but I think that it should also be mentioned in the Introduction.

Figure 1 B For the visualization of the injection, a superimposed figure with both the diagram and microphotograph could be useful, since the last is too dark to be appreciated.

Figure 2 E. What do the scattered lines mean?

Figure 3 B. is labelled as ChAT but I suppose it is GAD67

Figure 5. Confocal images are really nice.

213. Why only n=3 is used while in other parts it is stated n=6? 3 is a small sample size even being immunohistochemical assays…

Figure 7. Again, what do the dashed lines mean?

Author Response

Responses to reviewer 1:

nDB: Please use the standard abbreviations. The diagonal band is usually abreviated as DB or DBB (Diagonal band of Broca): Medial septum/diagonal band (MS/DB)

Response: According to your suggestion, the term “medial septum and nucleus of the diagonal band (MS/nDB)” has been changed to “medial septum/diagonal band (MS/DB)” throughout the manuscript, figures, and supplementary information.

Introduction: 50-58: It should be mentioned that the MS/DB is particularly involved in the generation of theta rhythm in the hippocampus, which is key for cognitive processing. That is well discussed further but I think that it should also be mentioned in the Introduction.

Response: We had already described regarding the role of MS/DB in the generation of theta rhythm in the hippocampus and cognitive processing in “Introduction” section (lines 48-59) without using the term “theta rhythm” but “neuronal oscillation”. According to your suggestion, we have added the term in order to make it further clear (lines 51-52).

Figure 1 B For the visualization of the injection, a superimposed figure with both the diagram and microphotograph could be useful, since the last is too dark to be appreciated.

Response: According to your suggestion, we have prepared a figure of the diagram that is overlaid by the microphotograph. The top figure has been remained unmodified. The explanation has been added to the figure legend (lines 128-129).

Figure 2 E. What do the scattered lines mean?

Response: In Figure 2E, the scattered lines depict the results at the days (day 2 and 3) when significant differences had been observed between groups, as referred in the figure legend (lines 138-139). Since the description only by the line graph (left) may be difficult to read the group differences, the bar graph (right) consisting of the results at the day 2 and 3 had been represented with marks (*) depicting significant differences.

Figure 3 B. is labelled as ChAT but I suppose it is GAD67

Response: You may refer to the labelling “ChAT” in Figure 4B and it was a mistake. That has been corrected and depicted as “GAD67”.

Figure 5. Confocal images are really nice.

Response: Thank you for your compliment.

  1. Why only n=3 is used while in other parts it is stated n=6? 3 is a small sample size even being immunohistochemical assays…

Response: Only in the analyses of Figure 5, we had used naïve mice. In the other figures, we used mice receiving the stereotaxic surgery. We had made efforts to minimize the number of animals, and predicted the localization of FABP3 between cholinergic and GABAergic neurons. It is not so different among naïve individuals. In addition, the purpose of this analysis is not to reveal the exact value of the ratio of FABP3-positive GABAergic neurons. Thus, we had not statistically quantified them. For these reasons, we had prepared a smaller sample size in this investigation compared to the others.

Figure 7. Again, what do the dashed lines mean?

Response: Like Figure 2E, the lines in Figure 7B (line graphs) depict the results at days when significant differences had been observed between groups, and the detailed results had been shown in Figure 7C (bar graphs with marks (* and #) depicting the significant differences). The explanation had been shown in the figure legend (lines 264-265).

Reviewer 2 Report

The manuscript by Matsuo et al. sets to investigate the role of alpha-synuclein (aSyn) and its interplay with FABP3 in causing cognitive decline in a mouse model of disease, especially focusing in the medial septum brain region. While I find the concept original, I believe that the manuscript could be further improved by better explaining the mechanism under investigation and justifying aspects of the experimental design. 

The hypothesis is that FABP3 mediates aSyn induced toxicity via LC-PUFAs or direct interaction between the two proteins; could the authors explain better the suggested mechanism/ hypothesis? According to the authors LC-PUFAs and/ or FABP3 are causing aSyn oligomerization, aggregation; is this observed in this particular study? Besides the accumulation of phosphorylated aSyn, are higher molecular weight species of aSyn or insoluble aSyn aggregates detected biochemically following the intrastriatal injections of aSyn fibrils in wild type and/ or Fabp3-/- mice? Also, how is the hypothesis of FABP3 binding to aSyn and therefore promoting aSyn aggregation, reconcile with the differential localization of the two proteins within the MS/nDB?

While it is interesting approach and novel design to inject bilaterally different strains of aSyn aggregates and monitor behavior over time there is very little discussion and conclusions on the effects of ribbons. It is not clear what was the purpose of testing different aSyn strains in context of this manuscript. Furthermore, while it seems that cognitive deficits become more pronounced at 60 days post injection (dpi), the immunohistochemical analysis was performed at the 30 dpi point. What was the purpose of the extended behavioral analysis to 60 dpi? I can't help but wonder if there is progressive GABAergic neuronal loss and further paSyn accumulation at a later timepoint.

Importantly, except for figure 5A showing uncropped images of Western blots none other supplemental material is available to download and review. Please provide all supplemental figures and tables that are referenced in the text.

Other edits: 

Figure 1: How many days after the injection does the figure in panel B, ATTO550, correspond to? Could you check comment "After the sterotaxic surgery, mice were divided for two experiments"? The mice should have been already assigned to two experiments even before the stereotactic surgeries since they are of different genotypes? 

Line 161: the following statement is not in agreement with the data in Fig. 4B: "aSyn fibrils significantly increased the number of GAD67-positive cells in MS/nB". Is the increase rather of paSyn-positive GAD67 cells?

Line 162: there is insufficient evidence to support that exogenously applied aSyn is intracellular, please rephrase.

Figure 4B: Left panel top row is mis-labelled as ChAT instead of GAD67?

Figure 5: Could you please indicate the age and gender of mice analysed for expression levels and localization of aSyn and FABP3?

Lines 241, 242: which are the referred aSyn strains? is aSyn monomer considered as "strain"?

Materials and methods, animals and stereotaxic surgery: please indicate gender of fabp3-/- mice. Also, could you please define the fabp3+/+ mice, were they C57BL/6 male age-matched littermates?

Line 362: Is not very clear to me, you have used different groups of animals for behavioral analyses at each timepoint, 30 and 60 dpi?

Line 432: is not clear what you mean as monomeric proteins. Have you used any filtration or similar method for excluding dimers, or higher order oligomers?

Finally, please proof read and correct typos such as the following: 

  • Line 28: a verb is missing? cognitive impairments [caused] by aSyn fibrils
  • Line 30: please rephrase, the sentence makes little sense
  • Line 40: Cognitive impairment (singular)
  • Line 75: a word missing in "FABP3 binds with aSyn to accumulate"?
  • Line 175: Please rephrase, the sentence makes little sense
  • Line 176: Please specify intrastriatal injection of what? aSyn fibrils?
  • Line 183: Please try to rephrase
  • Line 215: intrastriatal injection of aSyn fibrils in mice
  • Line 301: Please try to rephrase

Author Response

Responses to reviewer 2:

The hypothesis is that FABP3 mediates aSyn induced toxicity via LC-PUFAs or direct interaction between the two proteins; could the authors explain better the suggested mechanism/ hypothesis? According to the authors LC-PUFAs and/ or FABP3 are causing aSyn oligomerization, aggregation; is this observed in this particular study? Besides the accumulation of phosphorylated aSyn, are higher molecular weight species of aSyn or insoluble aSyn aggregates detected biochemically following the intrastriatal injections of aSyn fibrils in wild type and/ or Fabp3-/- mice? Also, how is the hypothesis of FABP3 binding to aSyn and therefore promoting aSyn aggregation, reconcile with the differential localization of the two proteins within the MS/nDB?

Response: Regarding the FABP3-mediated αSyn oligomerization/aggregation, we hypothesize that under pathological condition oxidative stress in synucleinopathies, FABP3 binds to αSyn and promotes its oligomerization/aggregation and causes its cytotoxicity. Although LC-PUFAs-induced αSyn pathology may also require binding to FABP3, FABP3 has much higher affinity for LC-PUFAs. These hypotheses have been demonstrated by not other groups but only our serial studies, which are shown in the manuscript with citations [33, 35, 36]. Our hypotheses are derived from the following observations:

1) FABP3 overexpression in cultured cells did not promote αSyn oligomerization without oxidative stress (such as MPP+ treatment), and when FABP3 is replaced by its mutated form without LC-PUFAs binding ability, arachidonic acid treatment did not cause MPP+-induced αSyn oligomerization and cell death ([33] Shioda et al., J. Biol. Chem. 2014).

2) The affinity of FABP3 for arachidonic acid is much higher compared to those of αSyn, indicating that LC-PUFAs (such as arachidonic acid) preferentially bind to FABP3 rather than αSyn under physiological conditions ([51-53] Richieri et al., J. Biol. Chem. 1994; Richieri et al., Biochemistry 2000; Golovko et al., Biochemistry 2006).

3) We also identified the binding site between αSyn and FABP3, and confirmed the conformational changes in αSyn in the presence of FABP3, though the manuscript is under revision.

4) A FABP3 inhibitor that competitively inhibits the LC-PUFAs binding to FABP3 completely inhibits FABP3/LC-PUFAs-mediated αSyn cytotoxicity ([35, 36] Yabuki et al., Int. J. Mol. Sci. 2020; Matsuo et al., Neuropharmacology 2019).

These hypotheses have been added to “Discussion” section (lines 275-287).

Regarding higher molecular weight or insoluble αSyn species, we have not investigated in a model of intrastriatal injection of αSyn fibrils, since we focused on investigation of the localization of pαSyn and FABP3 in the MS/DB. However, those αSyn species are also pathological features in synucleinopathies, so we should address the appearance of those species and the effect of Fabp3 deletion in future studies. We will investigate by biochemical analyses using non ATTO550-labelled αSyn assemblies because it is unclear whether there are any effects of the labelling on the recognition sites of αSyn by antibodies and the shifts of molecular weight. The statement has been added to “Discussion” section (lines 357-360).

While it is interesting approach and novel design to inject bilaterally different strains of aSyn aggregates and monitor behavior over time there is very little discussion and conclusions on the effects of ribbons. It is not clear what was the purpose of testing different aSyn strains in context of this manuscript. Furthermore, while it seems that cognitive deficits become more pronounced at 60 days post injection (dpi), the immunohistochemical analysis was performed at the 30 dpi point. What was the purpose of the extended behavioral analysis to 60 dpi? I can’t help but wonder if there is progressive GABAergic neuronal loss and further paSyn accumulation at a later timepoint.

Response: We had used αSyn ribbons in addition to fibrils because cognitive impairment caused by diffrent αSyn strains have not been reported, whereas motor function had been already compared between αSyn fibrils and ribbons (Peelaerts et al., 2015). There are some reports regarding differences in αSyn polymorphs (fibril and ribbon) in the brain between patients with synucleinopathies, which had been described in “Discussion” section (lines 289-293). For these reasons, we had investigated distinct effects of αSyn strains on cognitive function in this study. The explanation has been added to the first of “Results 2.1.” section to show the purpose (lines 93-94) and also in “Discussion” section (line 302-303).

              Only spatial memory assessed by Y-maze and the Barnes maze tasks was not impaired at 30 days even if αSyn fibrils had been injected. As one of the few studies focusing on cognitive decline in synucleinopathies, we had attempted to reveal when spatial memory becomes impaired in the model and had further assessed at 60 days. Although the cognitive impairment was observed at 60 days, it could not be concluded that the exacerbation is due to dysfunction only in the MS/DB and/or other regions such as hippocampus and cerebral cortex. One of the purposes of this study is to validate cognitive decline in mice model of intrastriatal αSyn injection and to reveal the relevance of earlier αSyn pathology in the MS/DB to cognitive decline but not temporal analyses of the relevance between cognitive decline and the pathological changes in the MS/DB. We had thus conducted behavioral analyses at 30 and 60 days while immunohistochemical ones only at 30 days.

Importantly, except for figure 5A showing uncropped images of Western blots none other supplemental material is available to download and review. Please provide all supplemental figures and tables that are referenced in the text.

Response: We have provided the supplemental files in the same format with the previous one (PDF) in addition to Microsoft Word files. If the PDF files could not be downloaded again, would you refer to the Word files?

Other edits:

Figure 1: How many days after the injection does the figure in panel B, ATTO550, correspond to? Could you check comment "After the sterotaxic surgery, mice were divided for two experiments"? The mice should have been already assigned to two experiments even before the stereotactic surgeries since they are of different genotypes?

Response: The images in Figure 1B had been acquired 30 days after the injection of αSyn fibrils. The explanation has been added to the figure legend (lines 128-129). Please note that the image has been overlaid on the diagram shown above, according to Reviewer 1. The sentence “After the stereotaxic surgery, mice were divided for two experiments.” was a mistake and we had tried to show that mice had been “subjected” to two experiments, and the division had been done before the injection. The explanation has been corrected in the figure legend (lines 121-122).

Line 161: the following statement is not in agreement with the data in Fig. 4B: "aSyn fibrils significantly increased the number of GAD67-positive cells in MS/nB". Is the increase rather of paSyn-positive GAD67 cells?

Response: As you mentioned, the sentence was a mistake and has been corrected to “increased the number of pαSyn-positive cells in GAD67-positive cells.” (lines 165-166).

Line 162: there is insufficient evidence to support that exogenously applied aSyn is intracellular, please rephrase.

Response: According to your suggestion, the sentence has been changed to “ATTO550-positive reactivities in the MS/DB seemed to be similar between αSyn strains.” (lines 166-167).

Figure 4B: Left panel top row is mis-labelled as ChAT instead of GAD67?

Response: The labelling “ChAT” in Figure 4B was a mistake. That has been corrected and depicted as “GAD67”.

Figure 5: Could you please indicate the age and gender of mice analysed for expression levels and localization of aSyn and FABP3?

Response: The samples for the analyses in Figure 5 had been prepared from male mice at the age of three months, in line with the age of the other mice when the stereotaxic surgery had been conducted. We have described the age and gender of mice in “Material and Methods” section (lines 434-435 and 456-457).

Lines 241, 242: which are the referred aSyn strains? is aSyn monomer considered as “strain”?

Response: The term “αSyn strains” contains all of the αSyn assemblies (monomer, fibril, and ribbon) used in this study in addition to PBS (as control for statistical analyses). Throughout the manuscript, the term represented the αSyn assemblies used in each experiment. For example, the term you mentioned (lines 244-245) refers to αSyn monomer and fibril, and PBS in Figure 7.

Materials and methods, animals and stereotaxic surgery: please indicate gender of fabp3-/- mice. Also, could you please define the fabp3+/+ mice, were they C57BL/6 male age-matched littermates?

Response: We used only male mice throughout the study including Fabp3-/- mice. Fabp3+/+ mice indicates C57BL/6 wild-type, age-matched, but not littermates. Fabp3+/+ and Fabp3-/- (C57BL/6 background) mice were separately crossed in each genotype and housed. The description of the definition of Fabp3+/+ mice, gender of Fabp3-/- mice, and age-matching have been added (lines 372 and 374-375).

Line 362: Is not very clear to me, you have used different groups of animals for behavioral analyses at each timepoint, 30 and 60 dpi?

Response: As you mentioned, we had used different mice for behavioral analyses between 30 and 60 days after the injection. The analyses regarding learning and memory are likely to depend on the retention memory, in particular Barnes maze and passive avoidance tasks in this study. If one animal is subjected to the same task at 30 and 60 days, the retention memory acquired at the former time could be a cue at the latter time. In addition, if cognitive functions between two mice are already different at 30 days, the extent to retain the memory could be also different when they are subjected to the same task at 60 days. Thus, in order to avoid the influence of the retention memory in the same task, we had used different mice for behavioral tasks between 30 and 60 days after the injection.

Line 432: is not clear what you mean as monomeric proteins. Have you used any filtration or similar method for excluding dimers, or higher order oligomers?

Response: In this sentence, we had tried to describe the standard protocol using SDS and heat-mediated denaturation. In order to avoid misunderstanding, the term “for monomeric proteins” has been deleted and the initial term “Extraction” has been changed to “Protein extraction” (line 455).

Finally, please proof read and correct typos such as the following:

Line 28: a verb is missing? cognitive impairments [caused] by aSyn fibrils

Response: The term has been corrected to “cognitive impairments caused by αSyn fibrils” (line 28).

Line 30: please rephrase, the sentence makes little sense

Response: In the last sentence you mentioned in “Abstract” section, we had tried to show that FABP3 mediates αSyn neurotoxicity in GABAergic neurons in the MS/DB so that FABP3 in this region could be a therapeutic target for demented patients with synucleinopathies. The sentence has been corrected (lines 28-31).

Line 40: Cognitive impairment (singular)

Response: The term “Cognitive impairments” has been corrected to “Cognitive impairment” including the parts other than line 41 if those describe general “cognitive impairment” (lines 28, 41, 82, 89, 123, 126, 324-325).

Line 75: a word missing in "FABP3 binds with aSyn to accumulate"?

Response: The term has been corrected to “FABP3 binds to αSyn and promotes its accumulation and oligomerization” (lines 76-77).

Line 175: Please rephrase, the sentence makes little sense

Response: The sentence “Previous studies suggest that neural projection and expression levels of endogenous αSyn could be determinant factors for the extent and vulnerability of αSyn spreading through the brain.” has been corrected as follows: Previous studies suggest that the extent of αSyn spreading and its cytotoxicity partly depend on neural projection and expression levels of endogenous αSyn (lines 178-179).

Line 176: Please specify intrastriatal injection of what? aSyn fibrils?

Response: The strains of αSyn (monomers and fibrils) have been added to the sentence (line 180).

Line 183: Please try to rephrase

Response: In the sentences you mentioned (lines 187-188), we had tried to show that expression levels of endogenous αSyn, one factor contributing to αSyn spreading and its cytotoxicity, have been remained unclear. The sentence has been corrected to “Expression levels of endogenous αSyn in the MS/DB should also be investigated as another factor that contributes to αSyn spreading and the toxicity.” (lines 187-188).

Line 215: intrastriatal injection of aSyn fibrils in mice

Response: According to your suggestion, we have corrected the term to “injection of αSyn fibrils” (line 218).

Line 301: Please try to rephrase

Response: The sentence “enhances the vulnerability to amyloid-beta to impair synaptic transmission and spatial memory in mice” has been corrected to “enhances the vulnerability to amyloid-beta, disrupts synaptic transmission, and impairs spatial memory in mice” (lines 318-319).

Round 2

Reviewer 2 Report

Thank you for addressing all comments, it all looks good now!